# Clinical and lifestyle patterns in Asian children with inflammatory bowel disease in the U.S.

**Wenjing Zong**[1], **Amit Patel**[2], **Vivian Chang**[1], **Elana B. Mitchel**[1], **Natalie Stoner**[3], **Robert N. Baldassano**[1], **Lindsey Albenberg**[1]*

1 Division of Gastroenterology, Hepatology, and Nutrition, Children's Hospital of Philadelphia, Philadelphia, Pennsylvania, United States of America, 2 Rowan University School of Osteopathic Medicine, Glassboro, New Jersey, United States of America, 3 Division of Clinical Nutrition, Children's Hospital of Philadelphia, Philadelphia, Pennsylvania, United States of America

* albenbergl@chop.edu

**Data Availability Statement:** All relevant data are within the paper and its Supporting information files.

## Abstract

### Background

While there are many epidemiologic studies of Asian immigrants to the West and risk of inflammatory bowel disease (IBD), the phenotype and lifestyle of Asian patients, particularly children, with IBD are not well described. In this study, we describe lifestyle practices, such as dietary pattern, as well as disease phenotype in Asian American children with IBD.

### Methods

We reviewed the records of children with IBD, ages 0 to 21 years old, and race identified as Asian, Indian, or Pacific Islander. Patients who received outpatient IBD care at our center between January 2013 and January 2020 were included. We excluded patients who were international second opinions, who did not have a definitive diagnosis of IBD, and in whom a diagnosis of IBD was made after 18 years of age. A survey, including a food frequency questionnaire adapted from NHANES DSQ with modifications to include culturally appropriate food elements, was designed and conducted within this cohort to assess for dietary patterns.

### Results

Asian patients in our cohort have similar phenotypes as non-Asians with few distinctive differences. There was a Crohn's disease and male predominance similar with non-Asians. However, there was a high rate of proctitis in ulcerative colitis in Asian patients. Asian patients reported a typical dietary pattern that reflects a Westernized pattern rather than a traditional pattern. Despite a similar dietary pattern, there was a high rate of 25-OH Vitamin D deficiency (44%) and insufficiency (40%).

### Conclusions

This single center study showed that the phenotype of Asian children with IBD in the U.S. is similar with that of non-Asian with a few distinct differences. The Asian children in our cohort

**Funding:** National Institutes of Health [T32 AI 118684-2] grant support to W.Z. https://www.niaid.nih.gov/grants-contracts/training-grants NASPGHAN Mentored Summer Research Program to A.P. https://naspghan.org/medical-student-resident-opportunities/ The funders had no role in study design, data collection and analysis, decision to publish, or preparation of the manuscript.

**Competing interests:** The authors have declared that no competing interests exist.

reported following a Westernized dietary pattern and lifestyle. However, there was a high rate of Vitamin D deficiency surrounding diagnosis, suggesting a need for vigilant monitoring.

## Introduction

Environmental factors are thought to play an important role in the recent rapid rise in incidence of inflammatory bowel disease (IBD) in traditionally low-risk regions of the world, such as South and East Asia [1, 2]. Similarly, immigrants moving from their low incidence native countries exhibit the high-incidence of the adopted country [3, 4]. While there are many epidemiologic studies of Asian immigrants to the West and risk of IBD, the phenotype and lifestyle of Asian patients, particularly children, are not well described. Herein, we describe early lifestyle, pre and post diagnosis dietary patterns, and phenotype in Asian American children with IBD.

The United States and Canada have seen an increase in IBD amongst minority groups who recently immigrated from low-risk regions. A US population-based study in Olmsted County, MN, reported that the incidence rates for whites and nonwhites increased by 39% and 134%, respectively, from 1970 through 2010 [5]. Although immigrants to Canada were found to have a lower incidence of IBD overall relative to non-immigrants, children of immigrants from certain regions had a similar risk of IBD as children of non-immigrants [6]. However, Asian pediatric IBD patients have a distinct phenotype from their non-Asian counterpart. Specifically, South Asian children in Canada with IBD have a male predominance, more complex Crohn's disease behavior such as perianal disease and more extensive colonic disease [7]. This clinical phenotype seems to be different from their native countries. For example, a recent large cohort study from India showed that perianal disease is rare in native Indian adults [8].

The aim of this study was to examine the phenotypic expression and lifestyle patterns in Asian children and adolescents with IBD at a large children's hospital. We hypothesized that Asian children with IBD living in the U.S. have a unique disease phenotype compared to non-Asians, and that they follow a lifestyle pattern that is acculturated to the West.

## Methods

### Patient population

Patients were initially selected by using a structured query language (SQL) algorithm of the electronic health record (EHR). The diagnosis of IBD was based on having a diagnosis code for IBD and an outpatient visit with a gastroenterologist provider at our pediatric center (S1 Data). Each chart was then retrospectively and independently reviewed carefully by two people, a gastroenterologist and a study team member for inclusion/exclusion criteria.

We included children with a diagnosis of IBD, ages 0 to 21 years old, report of race(s) that includes Asian, Indian, or Pacific Islander. Patients who received outpatient IBD care between January 2013 and January 2020 were included. We excluded patients who were international second opinions, who did not have a definitive diagnosis of IBD, and in whom a diagnosis of IBD was made after 18 years of age. Demographic information, IBD phenotype based on Paris classification [9, 10], clinical course, lifestyle factors such as dietary pattern were recorded.

Non-Asian pediatric IBD patients were selected as controls. Data was collected through the Improve Care Now (ICN) Registry at our center [improvecarenow.org]. ICN data was

available for 1301 patients who identified as White or African American only and actively being followed for their IBD from January 2017 through November 2019. Patients of mixed race or unknown race were excluded. Phenotypic information was collected from Healthy Planet smart-forms from the most recent clinical visit. Some phenotypic data was not available from the Healthy Planet smart forms and therefore not reported in the above table.

## Survey design

A dietary survey instrument was designed specifically for this study since no validated instrument was available in pediatric IBD patients and 24-hour dietary recalls were not appropriate for our goal of assessing changes in dietary pattern over the course of the disease. The food frequency questions were adapted from NHANES Dietary Screener Questionnaire (DSQ), a validated short form instrument that assess food frequency in subjections 2 to 60 years-old in the U.S. [11, 12]. Additional questions were developed to capture culturally appropriate dietary items such as fermented foods and traditional pattern of eating. Modifications were made to the DSQ to include Likert scales and photo representations of serving sizes for food elements due to ease of use by parental proxies. The minimal and maximal servings/day on the Likert scale were transformed into numerical values ranging from 0 to 100. Participants were asked about diet pre- and post-diagnosis, where food elements in typical diets 1–2 years prior to diagnosis and in current diet in the last 3 months were compared. Comparison between pre- and post- food consumption was done by paired T-test using the transformed numerical values. The food frequency instrument was reviewed with a registered dietician with expertise in IBD and a team of GI providers in our center prior to administration (S2 Data). Dietary acculturation was assessed through questions about dietary patterns at home, at school, outside of the home, use of traditional spices in cooking, and grocery shopping habits. The instrument was deployed as part of a survey on lifestyle practices through REDCap.

Individuals from the Asian cohort were invited via email/mail to complete the survey (S2 Data). All surveys were completed by parents, or patients if they were 18 years or older at the time of the survey. Parents were encouraged to corroborate with the patients when appropriate. Surveys were completed online through RedCap. If no response, subjects and parents completed the surveys at the time of a clinic visit, infusion visit, hospitalization, or procedure visit. Individuals were excluded from the survey data analysis if Asian race was not confirmed or if they did not complete the survey.

## Statistical analysis

Study data was collected and managed using REDCap, a secure, web-based application designed to support data capture for research studies. Standard descriptive statistics were used to describe baseline characteristics including means, standard deviations, medians, and ranges. Differences in proportions of CD, complex CD behavior (stricturing and/or penetrating disease), perianal disease, and extent of UC between Asians and African Americans, between Asians and Caucasians, and between African Americans and Caucasians were assessed using hypothesis testing with 95% confidence interval. Linear regression was used to evaluate the association between vitamin D level and markers of inflammation such as C-reactive protein, sedimentation rate, and albumin. For all analyses, a p value of $<0.05$ was considered significant. Average change in Likert scale scoring was evaluated among participants pre- and post-diagnosis for each food group in the dietary frequency questionnaire. All analyses were performed using Stata 15.1 software (StataCorp. 2018. *Stata Statistical Software: Release 15.1*. College Station, TX: StataCorp LP). Biostatics support was received through the Biostatistics and Data Management Core at the Research Institute at our institution.

## Ethical considerations

All survey participants and/or legal guardians provided informed consent, as well as assent, if applicable. The study protocol was approved by the Institutional Review Board (IRB) at the Children's Hospital of Philadelphia. Informed consent was waived by the IRB for the retrospective portion of the study. Informed consent was obtained for the survey portion of the study.

## Results

### Demographics

In the retrospective portion of the study, 158 patients met inclusion criteria. Six patients were excluded—4 due to incomplete record and inability to confirm diagnosis, and 2 for identifying their country of origin that is outside of the Census Bureau's definition of Asian race. The survey was sent to 152 subjects and 86 were completed. Seven surveys were excluded due to incomplete data. Seventy-nine surveys were analyzed.

The median age of diagnosis was 11 years (IQR 6, 14) in the Asian cohort. There was a male predominance (57%). Ninety-six patients (63%) were diagnosed with Crohn's disease (CD), 37 (24%) with ulcerative colitis (UC), and 19 (13%) with indeterminate colitis (IC). The control cohort consisted of 1301 patients, of which 747 (57%) were male. There were 156 patients identified as African American and 1145 identified as Caucasian. The proportions of each IBD subtype were not statistically different from our Asian cohort (Table 1).

The Asian IC subgroup had a lower median age of diagnosis at 6 years old (IQR 4–15) compared to Asian UC or CD subgroup, potentially reflecting the pan-colitis phenotype of many early onset IBD presentations and the challenge of making a definitive diagnosis in younger children since clinical phenotype can evolve over time [13]. A relatively large percentage of 19% (n = 29) of the Asian cohort were diagnosed before the age of 6 years. While this may reflect the referral pattern for our quaternary children's hospital, zip code data showed that 92% of the Asian patients included in our cohort reside locally within PA where our center is located, and also in nearby states that our center reaches (NY, NJ, DE).

### Disease phenotype

In the Asian CD cohort, 20% had perianal phenotype. This is similar to the 17% in Caucasians from our center. The percentage of CD patients with complex disease behaviors such as stricturing and penetrating disease were similar with Caucasians, whereas African American patients in our center had higher proportion with complex behavior. In the Asian UC subgroup, majority of the patients had pancolitis, though there was a significantly larger percentage of E1 disease, or proctitis (22%) compared with Caucasians (Table 1). Overall, a small percentage of the Asian cohort (5%) developed extra-intestinal manifestations as part of their IBD.

Within the Asian cohort, 108 identified as South Asian (S1 Table). There was a CD (66%) and male (58%) predominance in this subgroup, similar to the full Asian cohort. Within CD, 23% had perianal disease. Within UC, 20% patients had proctitis.

### Family history and acculturation

Of the 79 surveys analyzed, 54% reported country of origin to be India, 9% Pakistan, 6% China, 6% Korea and 20% others. 85% of the surveyed patients were born in the U.S., of whom 75% were second generation immigrants. Of the 56 subjects reported to be South Asians on the survey, all but 9 were born in the U.S. Eleven out of 79 (14%) Asian subjects reported

**Table 1. Baseline characteristics of Asian American cohort vs. a control cohort of non-Asian (African American and Caucasian) patients from our center's ICN database.**

|  | **Asian** N = 152 | **African American** N = 156 | **Caucasian** N = 1145 | **P value** |
|---|---|---|---|---|
| **IBD type** n (%) |  |  |  |  |
| CD | 96 (63) | 112 (75) | 796 (70) | 0.1*, 0.11+ |
| UC | 37 (24) | 31 (20) | 218 (19) | 0.34*, 0.12+ |
| IBD-U | 19 (13) | 13 (8) | 131 (11) | 0.23*, 0.7+ |
| **Sex: Male** n (%) | 87 (57) | 90 (58) | 656 (57) | 0.94*, 0.99+ |
| **Crohn's disease** | **Asian** | **African American** | **Caucasian** | **P value** |
| Location, n (%) | N = 96 | N = 106** | N = 777** |  |
| L1 (ileal) | 9 (9%) | 12 (11%) | 104 (14%) |  |
| L2 (colonic) | 22 (23%) | 17 (16%) | 129 (17%) |  |
| L3 (ileocolonic) | 62 (65%) | 73 (69%) | 524 (67%) |  |
| L4a (upper disease proximal to LOT) | 15 (16%) | 30 (28%) | 209 (27%) |  |
| L4b (upper disease distal to LOT) | 2 (2%) | 14 (13%) | 88 (11%) |  |
| L4ab | 3 (3%) | 14 (13%) | 111 (14%) |  |
| Behavior, n (%) | N = 96 | N = 110** | N = 783** |  |
| B1 (non-stricturing, non-penetrating) | 80 (83%) | 67 (61%) | 621 (79%) |  |
| B2 (stricturing) | 9 (9%) | 23 (21%) | 89 (11%) |  |
| B3 (penetrating) | 2 (2%) | 10 (9%) | 39 (5%) |  |
| B2B3 (stricturing, penetrating) | 5 (10%) | 10 (9%) | 34 (4%) |  |
| Complex behavior (stricture and/or penetrating disease) |  |  |  | <0.05*, 0.34+ |
| Perianal | 19 (20%) | 32 (29%) | 132 (17%) | 0.08*, 0.49+ |
| **Ulcerative colitis** | **Asian** | **African American** | **Caucasian** | **P value** |
| Extent, n (%) | N = 37 | N = 31 | N = 238 |  |
| E1 (proctitis) | 8 (22%) | 2 (6%) | 14 (6%) | 0.09*, 0.05+ |
| E2 (left side; distal to splenic flexure) | 3 (8%) | 10 (32%) | 43 (18%) | 0.01*, 0.13+ |
| E3 (extensive; distal to hepatic flexure) | 5 (14%) | 1 (3%) | 21 (9%) | 0.14*, 0.36+ |
| E4 (pancolonic) | 21 (57%) | 18 (58%) | 160 (67%) | 0.91*, 0.21+ |

*Comparison between A vs AA

+Comparison between A vs C

**Few missing data from ICN Healthy Planet smart forms

family history of IBD in a first degree relative and 16 out of 79 (20%) in a second degree relative. Twenty-three patients (29%) of the Asian subjects surveyed reported any family history of IBD.

We assessed for acculturation through a series of questions in the survey (S2 Data). Traditionally preserved foods such as pickled mango, kimchi, were rarely consumed. However, traditional spices such as cumin, saffron, peppercorn, were used in home cooking about several times weekly. Asian patients tended to consume Westernized foods more than traditional foods for breakfast and for meals outside of the home (Fig 1). They were slightly more likely to consume a traditional meal while at home.

## Dietary pattern pre- and post-diagnosis

We assessed the dietary pattern pre- and post-diagnosis of IBD. Consumption of soft drinks, artificial coloring, sweeteners, preservatives, and emulsifiers were reported to be lower than 1 serving per day at baseline before diagnosis of IBD (Fig 2).

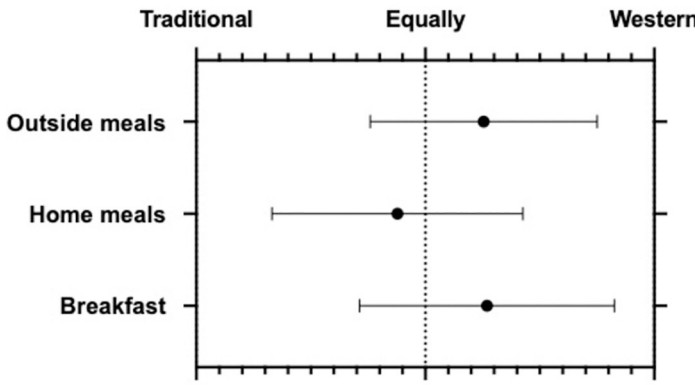

**Fig 1. Meal pattern in Asian children with IBD.** Meals trend toward a Westernized pattern, especially for outside meals and breakfast.

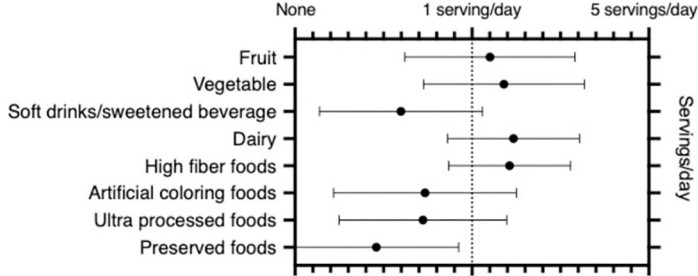

**Fig 2. Pre-diagnosis dietary pattern.** Pre-diagnosis dietary pattern shows a Westernized pattern of intake with lower amount of fiber, vegetables, and fruits.

Additionally, we were interested in the reported change in dietary pattern after diagnosis of IBD. We found a change in frequency of consumption of several food elements post-diagnosis (Fig 3). Amount of each food element pre- and post-diagnosis for each patient was compared. Consumption of fruit and vegetable remained the same after IBD diagnosis and around 1 serving a day. Intake of dairy decreased after diagnosis from an average of 2 servings/day to 1 servings/day. About 2/3 of all patients reduced their dairy intake post-diagnosis. Overall intake of sweetened beverages and soda was low pre-diagnosis, at less than 1 cup/day, and patients reduced consumption even more post-diagnosis. Consumption of highly processed foods occurred a few times per week before diagnosis, and there was a trend towards decreasing frequency post diagnosis. Consumption of foods with high amounts of emulsifiers and additives decreased after diagnosis, from 1 serving/day to 0.5 servings/day. About 3/4 of our cohort reported a decrease in the frequency of consumption of emulsifiers and food additives post-diagnosis.

## Vitamin D status

We examined the 25-OH Vitamin D serum levels in our Asian cohort. We define deficiency as a level <20 ng/ml (50 nmol/L), and insufficiency as a level between 21–29 ng/ml (52.5–72.5 nmol/L) [14]. Our IBD center's clinical practice is to recommend standard vitamin D intake from vitamin D containing foods or supplements per American Academy of Pediatrics (AAP) guidelines. Therefore, patients are typically not started on any repletion therapy of vitamin D

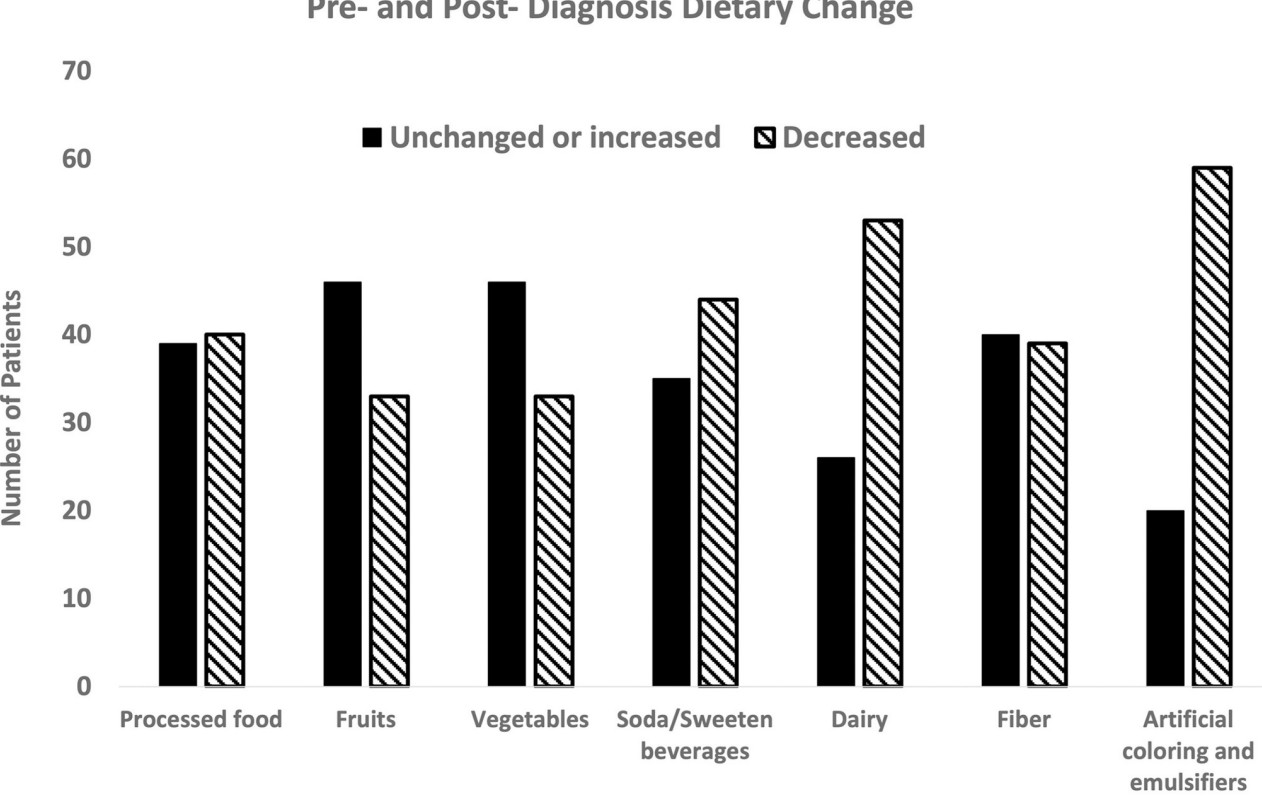

**Fig 3. Pre- and post- diagnosis dietary changes.** Post-diagnosis dietary patterns show a decrease in consumption of dairy, soft drinks, and foods with artificial coloring/emulsifiers when compared pre-diagnosis patterns. There is an increase in fruits and vegetable consumption post-diagnosis.

unless directed by the level. We examined levels obtained closest to IBD diagnosis as available, with most of the levels were obtained within 1 year of diagnosis (median 1.2 years). There was a high percentage with vitamin D deficiency (44%) and insufficiency (40%) (Fig 4). Using linear regression, CRP was found to be inversely associated with Vitamin D level (-0.67, CI -1.34, -0.0006, p<0.05), but the other covariates were not significant including albumin and ESR (S1 Fig). The levels were also not associated with the time of the year at which the level was drawn (S1 Fig).

## Discussion

IBD is increasing among immigrants from Asia to Western countries. Children of immigrants often assume the higher incidence risk as children of non-immigrants [15]. While the Asian immigrant cohort consists of people with differing genetic backgrounds and cultural practices, this trend suggests that lifestyle and environmental exposures may be common factors that activate the underlying risk to developing IBD. In this study conducted in a large U.S. children's hospital, the cohort of pediatric Asian subjects consists mostly of second generation South Asian children. Asian patients have similar phenotypes as non-Asians with few distinctive differences, specifically in the rate of perianal disease in those with CD in South Asian patients. This is consistent with findings from South Asian pediatric patients in Canada as well as in several adult studies [16–19]. Of interest, adult U.S. born Asians in a multicenter study had higher rate of perianal disease but relatively lower rate of perianal procedures. Further

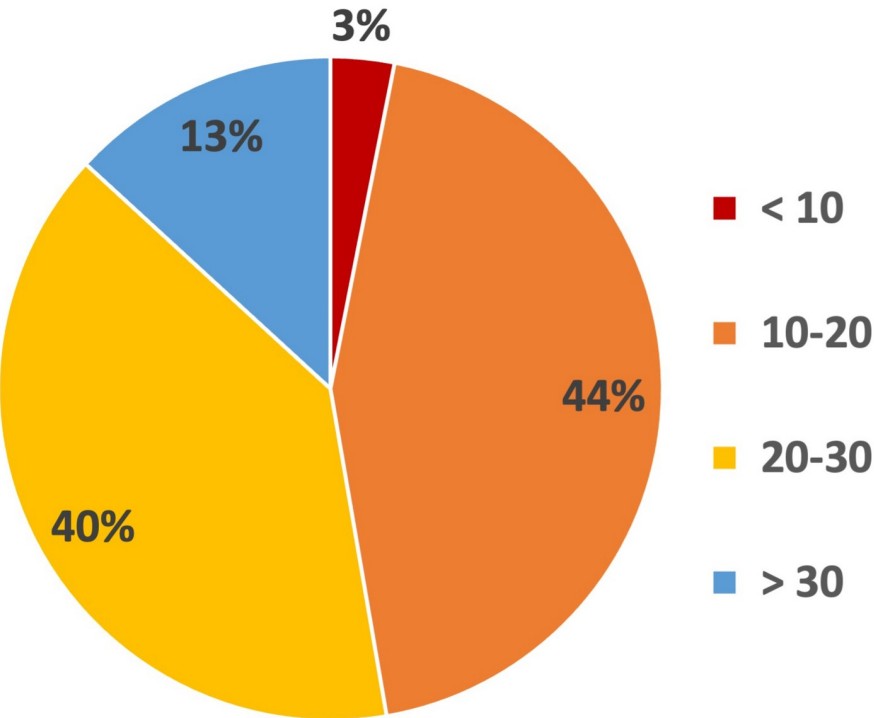

**Fig 4. Serum 25-OH Vitamin D levels around diagnosis.** Vitamin D deficiency and insufficiency rates are high in this cohort.

studies are needed in pediatrics to understand perianal disease course and potential genetic markers in Asians.

We then examined the lifestyle practices in our Asian cohort and hypothesize that they would follow a Westernized pattern. Much of our cohort consists of second-generation immigrant children from South Asia. Indeed, we found the baseline pre-diagnosis dietary pattern in general mimics that of a Western pattern of consumption instead of the traditional and cultural diet of their family. The dietary pattern in the U.S. per NHANES generally include an adequate intake of dairy in toddlers that decreases over time, with modest intake of fiber, vegetables, and fruits [20–26]. While they do consume vegetables and fruits daily, the number of servings per day is significantly less than recommended [27]. Similar to pediatric patients with CD in another study, our cohort consistently consume some processed foods as well as foods containing artificial ingredients [28]. Many Asian cultures include preserved or fermented foods as part of their cuisine, such as kimchi, pickled mango, natto, etc [29]. These foods are rich sources of microbes and are hypothesized to provide benefit in the intestinal microbiome. However, we found that the Asian children in our cohort rarely consumed these traditional foods. Together, these findings suggest that Asian children with IBD have similar dietary patterns as non-Asians, which in part may explain the similar phenotype with few differences. Of interest, we assessed the self-reported dietary pattern after diagnosis of IBD and the only category in which there was a clinically significant change was a decrease in consumption of foods with high amounts of emulsifiers and additives. This likely reflects the increased recognition of the role of diet in IBD by our patients' families. Together with a family history rate similar to a

cohort of pediatric IBD patients in Israel [30], our findings raise suspicion for an interaction between lifestyle practices on top of an underlying risk background in the development of IBD in Asian children in the U.S.

Our Asian cohort had a high proportion of patients with Vitamin D deficiency with levels <20 ng/ml (44%). A historic IBD cohort with mostly Caucasian and African American children from our center showed 16% of the CD patients with serum level less than 15 ng/mL (38 nmol/L) [31], though this comparison is affected by a slightly different definition of deficiency as well as focus on the CD subgroup only. A large cohort of healthy U.S. children in NHANES showed that 9% were 25-OH Vitamin D deficient and 61% insufficient [32]. Furthermore, the mean intake of Vitamin D and calcium in diets of Asian children 1–6 years in NHANES were not significantly different from non-Hispanic White children [20]. We observed that our Asian cohort followed a mostly Westernized pattern of intake, though seemed to have a high rate of Vitamin D deficiency surrounding diagnosis. A variety of factors may contribute to this observation, including disease activity, pigmentation of skin and different genetic polymorphisms in this population [33]. Our regression analysis revealed inverse correlation with CRP in our cohort but lacked the number to support association with other markers of disease activity. The retrospective nature of this study also could not delineate between various underlying mechanisms. However, this warrants special attention for screening since serum vitamin D level is associated with clinical outcomes in IBD [34, 35]. Of particular interest will be whether this alters important outcomes in the pediatric IBD population, such as bone health.

Our study has several limitations. While the single-center nature of the study allows for generally consistent care by physicians in our IBD center, it is limited by selection bias. And despite a relatively large number of Asian patients, this may not reflect the overall trend in the country since most of our patients are from the local region our center serves (Pennsylvania, New Jersey, and New York) and may be skewed by a small percentage of second opinion patients from other states in the U.S. (We excluded international second opinions). Our cohort also consists of a mixture of children from different regions in Asia with very different genetic and cultural backgrounds which may not be generalizable. Given the large South Asian population and specific phenotypic expression based on other studies, we characterized the South Asian separate from the full Asian cohort, but there may be difference in phenotype and lifestyle practices within regions in South Asia. Additionally, the retrospective nature of the phenotype portion is limited by recall bias. And finally, there was a difference in collection of retrospective data between the Asian cohort and the control cohort, which may have affected the comparison. The control cohort data was collected from Healthy Planet smart forms that is part of our center's ICN registry, whereas the Asian cohort data was collected manually through chart review. This was done because we were interested in more granular data within the Asian cohort that may not be otherwise available through ICN registry. Our survey response rate was only 52% and may have led to selection bias. Survey instruments are inherently limited by recall bias and measurement bias. To the best of our ability, we tried to decrease measurement bias by adapting a validated self-report instrument such as the NHANES DSQ within our survey. We also included photo representation of serving sizes and food items in question to aid recall in the absence of trained interviewers.

In summary, this study confirmed a similar phenotypic presentation of IBD with a few unique distinctions in Asian children compared with non-Asians in the U.S. It also showed a Westernized dietary pattern and lifestyle in our Asian cohort. Despite a similar dietary pattern, these children have high rate of vitamin D deficiency, suggesting a need for vigilant monitoring.

## Supporting information

**S1 Fig. Analysis of Vitamin D level.** Regression models show that vitamin D levels and CRP are inversely correlated, but not for other markers of inflammation. Vitamin D levels have no significant correlation with time of the year in our cohort.
(PPTX)

**S1 Data. ICD-9 IBD diagnosis codes.**
(DOCX)

**S2 Data. RedCap survey design.**
(PDF)

**S1 Table. Baseline characteristics of South Asian American patients.**
(DOCX)

## Acknowledgments

We would also like to thank the generous IBD families as well as CURE for IBD for their support of the CHOP IBD Center.

## Author Contributions

**Conceptualization:** Wenjing Zong, Natalie Stoner, Robert N. Baldassano, Lindsey Albenberg.

**Data curation:** Wenjing Zong, Amit Patel, Vivian Chang, Natalie Stoner.

**Formal analysis:** Wenjing Zong, Amit Patel, Elana B. Mitchel, Natalie Stoner.

**Funding acquisition:** Wenjing Zong.

**Methodology:** Wenjing Zong, Natalie Stoner, Lindsey Albenberg.

**Project administration:** Wenjing Zong, Vivian Chang.

**Supervision:** Wenjing Zong, Lindsey Albenberg.

**Writing – original draft:** Wenjing Zong, Amit Patel, Vivian Chang.

**Writing – review & editing:** Wenjing Zong, Amit Patel, Vivian Chang, Elana B. Mitchel, Natalie Stoner, Robert N. Baldassano, Lindsey Albenberg.

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
