## [Decision Letter · Decision Letter 0]

31 Oct 2022

PONE-D-22-24143CLINICAL AND LIFESTYLE PATTERNS IN ASIAN CHILDREN WITH INFLAMMATORY BOWEL DISEASE IN THE U.S.PLOS ONE

Dear Dr. Albenberg,

Thank you for submitting your manuscript to PLOS ONE. After careful consideration, we feel that it has merit but does not fully meet PLOS ONE’s publication criteria as it currently stands. Therefore, we invite you to submit a revised version of the manuscript that addresses the points raised during the review process.

We look forward to receiving your revised manuscript.

Kind regards,

Valérie Pittet, PhD

Academic Editor

PLOS ONE

Journal Requirements:

2. In the online submission form and the Methods section of your manuscript, please amend your current ethics statement to include the full name of the ethics committee/institutional review board(s) that approved your specific study.

3. In the ethics statement in the manuscript and in the online submission form, please provide additional information about the patient records/samples used in the retrospective portion of your study. Specifically, please ensure that you have discussed whether all data/samples were fully anonymized before you accessed them and/or whether the IRB or ethics committee waived the requirement for informed consent. If patients provided informed written consent to have data/samples from their medical records used in research, please include this information.

5. Please ensure that you include a title page within your main document. You should list all authors and all affiliations as per our author instructions and clearly indicate the corresponding author.

8. We note that Supplemental Figure 2 in your submission contain copyrighted images. All PLOS content is published under the Creative Commons Attribution License (CC BY 4.0), which means that the manuscript, images, and Supporting Information files will be freely available online, and any third party is permitted to access, download, copy, distribute, and use these materials in any way, even commercially, with proper attribution. For more information, see our copyright guidelines: http://journals.plos.org/plosone/s/licenses-and-copyright.

a. You may seek permission from the original copyright holder of Supplemental Figure 2 to publish the content specifically under the CC BY 4.0 license. 

Reviewers' comments:

Reviewer's Responses to Questions

**Comments to the Author**

1. Is the manuscript technically sound, and do the data support the conclusions?

Reviewer #1: No

Reviewer #2: Partly

2. Has the statistical analysis been performed appropriately and rigorously? 

Reviewer #1: No

Reviewer #2: Yes

3. Have the authors made all data underlying the findings in their manuscript fully available?

Reviewer #1: Yes

Reviewer #2: Yes

4. Is the manuscript presented in an intelligible fashion and written in standard English?

Reviewer #1: Yes

Reviewer #2: Yes

5. Review Comments to the Author

Reviewer #1: In this study, Wenjing Zong, et al. described disease phenotype and lifestyle practices including dietary pattern in Asian American children with Inflammatory Bowel Disease.

Strengths of the study:

- Study question is valid

- Adequate literature review was performed.

- The author's major findings were clearly presented. They adequately address the stated research objectives.

-

The manuscript can be improved by addressing following concerns.

- Research methodology is not clearly explained. How did authors compare dietary patterns between the study population and control group? It is not clear if the difference between the two groups is statistically significant. It is difficult to conclude that the study population followed a Westernized diet that increased the risk of developing IBD based on the study findings.

- Authors may want to add hospitalizations and Surgical history data for study/control population

- Correct Spelling mistake - About 2/3 of all patients reduced diary intake post-diagnosis (Results section Line 196).

Reviewer #2: The authors of this manuscript seek to define the phenotype of children with IBD who are of Asian ancestry here in the U.S. They describe a moderate size cohort of patients amidst a large quarternary care center. Although the phenotype appears similar to other patients seen at the center, they report a higher rate of perianal involvement in their Asian patients with Crohn's disease and higher rates of proctitis in their Asian patients with UC. They go on to describe dietary habits (pre- and post-diagnosis).

This is an important topic (that is appropriate for the audience of PLoS One) although I do have some concerns about the manuscript as currently written including a number of clarifications that are needed as it takes away from the flow of the paper.

Major Concerns:

1) The duration of IBD diagnosis is not described. The authors allude to the recall bias limitation but that may be magnified if patients were diagnosed many years prior to the survey as compared to a recent diagnosis. This would be especially true for an adult-aged patient who won't be

2) Some of the comparative data was extracted from a worksheet via the ICN pathway suggesting that procedure to collect/confirm phenotypic data on cases whereas the non-Asian IBD cohort was collected in a different process. This discrepancy should at least be addressed in the limitations if not corrected

3) At the end of the Survey Design section in Methods, it is stated that if the survey failed to confirm whether the race of the subject is race, they would be excluded and they also would be excluded if they did not complete the survey. It is not clear if this exclusion is for only the survey section or for the entire study. I would assume that with only 79 surveys analyzed, that this pertains only to the dietary analysis. This should be clarified.

4) The analysis regarding Vitamin D status found that time of year was not associated with level. The description of Vitamin D was not discussed well in the methods section. It is not clear if multiple Vitamin D levels from the same patient was included in the analysis or if random values of Vitamin D across individual patients were compared. Similarly, it's not clear if included levels were after repletement.

5) It's not clear if the determination of whether someone had Vitamin D deficiency or insufficiency were through the entirety of their Crohn's history or only in one particular time course. While the numbers are high, if someone who had sufficient levels in Years 1, 2, and 4 had a low Vitamin D level in Year 3, that would be a different phenotype than someone deficient in Years 1-4. This needs to be clarified. Reference 32 also uses a different definition of Vitamin D deficiency so the 16% cited isn't an adequate comparator.

6) Much of the dietary data is descriptive in nature without clear statistical analysis but statements are made about certain foods being consumed at rare rates. Is there any strict definition of "rare"?

Minor Concerns:

1) It's not fully clear to me what is meant in the patient population section that the charts of cases were reviewed by "a gastroenterologist and a study team member" (i.e. whether 2 people reviewed it or whether it was a study team member that was a gastroenterologist).

2) It's not fully clear to me why international second opinions were excluded but other second opinions remained in (as their phenotype might be less accurate)

3) When the age at diagnosis is discussed for the Asian patients with IC, it is stated that it is lower, although it's not clear what comparative group is being used. I suspect this is "lower than Asian UC cohort and Asian CD cohort" but that should be clarified. Similarly, in that section, there is the statement of a large percentage of patients diagnosed before 6 years of age but not clear if this is the Asian IBD group or Asian IC group (or another group).

4) When zip code data was analyzed and 92% of patients were reported to reside "locally", it's not clear what is meant: within state? within several states? by geographic distance?

5) In the Asian UC cohort, they describe that the "highest proportion of the patients have pancolitis". I suspect this is meant to be the majority of patients have pancolitis rather than this cohort has a higher percentage of pancolitis (since Table 1 suggests that is not the case).

6) The phenotype of the South Asian patients seems to be quite similar. It seems that this section could be shortened and that table be moved in as a Supplementary Table. The wording around the perianal Crohn's involvement gives the impression that this is different than the overall Asian cohort but it appears that it is quite similar.

7) It might be helpful to also state what percentage of patients have EITHER a first or second degree relative with IBD. It's not clear whether the 16 out of 79 patients that had a second degree family member with IBD did NOT have a first degree family member with IBD.

8) There appears to be a word missing in the paragraph in the Discussion section regarding Vitamin D status that starts with "Furthermore"

6. PLOS authors have the option to publish the peer review history of their article (what does this mean?). If published, this will include your full peer review and any attached files.

Reviewer #1: No

Reviewer #2: No

---

## [Author Response · Author response to Decision Letter 0]

12 Jan 2023

Thank you to the editors and reviewers for time and effort. We value these constructive feedbacks to improve our manuscript. We have responded to each comment as below and updated the manuscript accordingly.

Updated title page and manuscript to meet style requirement.

2. In the online submission form and the Methods section of your manuscript, please amend your current ethics statement to include the full name of the ethics committee/institutional review board(s) that approved your specific study.

Updated CHOP IRB in Methods section. 

3. In the ethics statement in the manuscript and in the online submission form, please provide additional information about the patient records/samples used in the retrospective portion of your study. Specifically, please ensure that you have discussed whether all data/samples were fully anonymized before you accessed them and/or whether the IRB or ethics committee waived the requirement for informed consent. If patients provided informed written consent to have data/samples from their medical records used in research, please include this information.

Updated in ethics statement in manuscript. 

Updated ORCID ID.

5. Please ensure that you include a title page within your main document. You should list all authors and all affiliations as per our author instructions and clearly indicate the corresponding author.

Included title page within manuscript document. Added indication for corresponding author and updated affiliation per instructions.

This was updated as above.

Updated supporting information captions and in-text citations.

8. We note that Supplemental Figure 2 in your submission contain copyrighted images. All PLOS content is published under the Creative Commons Attribution License (CC BY 4.0), which means that the manuscript, images, and Supporting Information files will be freely available online, and any third party is permitted to access, download, copy, distribute, and use these materials in any way, even commercially, with proper attribution. For more information, see our copyright guidelines: http://journals.plos.org/plosone/s/licenses-and-copyright.

a. You may seek permission from the original copyright holder of Supplemental Figure 2 to publish the content specifically under the CC BY 4.0 license. 

Updated S2 Fig to remove copyrighted images.

Comments to the Author

1. Is the manuscript technically sound, and do the data support the conclusions?

Reviewer #1: No

Reviewer #2: Partly

2. Has the statistical analysis been performed appropriately and rigorously?

Reviewer #1: No

Reviewer #2: Yes

3. Have the authors made all data underlying the findings in their manuscript fully available?

Reviewer #1: Yes

Reviewer #2: Yes

4. Is the manuscript presented in an intelligible fashion and written in standard English?

Reviewer #1: Yes

Reviewer #2: Yes

5. Review Comments to the Author

Reviewer #1: In this study, Wenjing Zong, et al. described disease phenotype and lifestyle practices including dietary pattern in Asian American children with Inflammatory Bowel Disease.

Strengths of the study:

- Study question is valid

- Adequate literature review was performed.

- The author's major findings were clearly presented. They adequately address the stated research objectives.

-

The manuscript can be improved by addressing following concerns.

- Research methodology is not clearly explained. How did authors compare dietary patterns between the study population and control group? It is not clear if the difference between the two groups is statistically significant. It is difficult to conclude that the study population followed a Westernized diet that increased the risk of developing IBD based on the study findings.

The dietary pattern survey was only administered to the study population. We added a sentence to clarify this in the Methods section in the manuscript. We were specifically interested in the Western vs traditional pattern of intake in the study population before and after their IBD diagnosis. We did not have resources to conduct the same survey in control group, though we are very interested to gather this information in the control group and compare to study population as the next step. Given that we did not have this comparison and that this is a retrospective study with a prospective survey arm, we did not seek to conclude that a Westernized diet increased the risk of developing IBD. We sought to describe the dietary pattern in the study population and to assess any immediate nutritional considerations. Further studies looking into causal relationship between diet and incidence of IBD in Asian American children would be interesting and necessary.

- Authors may want to add hospitalizations and Surgical history data for study/control population

We initially did wish to look at hospitalization and surgery data for both populations as these are important clinical outcomes to understand. However, given the incomplete nature of the data from retrospective chart review, we were unable to make definitive associations. A larger study that may overcome subjects with incomplete/missing data or a prospective study that carefully records hospitalization/surgery rates would be better in answering this very important question.

- Correct Spelling mistake - About 2/3 of all patients reduced diary intake post-diagnosis (Results section Line 196).

Corrected spelling mistake.

Reviewer #2: The authors of this manuscript seek to define the phenotype of children with IBD who are of Asian ancestry here in the U.S. They describe a moderate size cohort of patients amidst a large quarternary care center. Although the phenotype appears similar to other patients seen at the center, they report a higher rate of perianal involvement in their Asian patients with Crohn's disease and higher rates of proctitis in their Asian patients with UC. They go on to describe dietary habits (pre- and post-diagnosis).

This is an important topic (that is appropriate for the audience of PLoS One) although I do have some concerns about the manuscript as currently written including a number of clarifications that are needed as it takes away from the flow of the paper.

Major Concerns:

1) The duration of IBD diagnosis is not described. The authors allude to the recall bias limitation but that may be magnified if patients were diagnosed many years prior to the survey as compared to a recent diagnosis. This would be especially true for an adult-aged patient who won't be

This is a very valid concern and unfortunately, we did not collect duration of IBD diagnosis in this study. The median age of diagnosis in our study cohort was 11 years, with IQR 6-14, so fortunately most of the cohort was not adult-aged patients and the recall would be less in duration. Given some of the interesting questions that came up with this pilot study, our next step could be to collect information regarding the duration of diagnosis and to stratify by recent vs distant diagnosis in a larger cohort.

2) Some of the comparative data was extracted from a worksheet via the ICN pathway suggesting that procedure to collect/confirm phenotypic data on cases whereas the non-Asian IBD cohort was collected in a different process. This discrepancy should at least be addressed in the limitations if not corrected

Thank you for pointing this out, we have added this limitation to the manuscript.

3) At the end of the Survey Design section in Methods, it is stated that if the survey failed to confirm whether the race of the subject is race, they would be excluded and they also would be excluded if they did not complete the survey. It is not clear if this exclusion is for only the survey section or for the entire study. I would assume that with only 79 surveys analyzed, that this pertains only to the dietary analysis. This should be clarified.

We have updated the survey design section in methods to say that this pertains only to the survey analysis.

4) The analysis regarding Vitamin D status found that time of year was not associated with level. The description of Vitamin D was not discussed well in the methods section. It is not clear if multiple Vitamin D levels from the same patient was included in the analysis or if random values of Vitamin D across individual patients were compared. Similarly, it's not clear if included levels were after repletement.

We added clarifications to the Vitamin D section to say that the levels we analyzed are obtained closest to the date of diagnosis as available. Patients are typically on standard recommendations for vitamin D intake per AAP, and therefore unlikely to be on repletion therapy until it is indicated by the level. However, given some time has elapsed between diagnosis (therefore initiation of IBD therapy) and some of the recorded Vitamin D levels, we cannot control for improvement of levels after initiation of IBD therapy.

5) It's not clear if the determination of whether someone had Vitamin D deficiency or insufficiency were through the entirety of their Crohn's history or only in one particular time course. While the numbers are high, if someone who had sufficient levels in Years 1, 2, and 4 had a low Vitamin D level in Year 3, that would be a different phenotype than someone deficient in Years 1-4. This needs to be clarified. Reference 32 also uses a different definition of Vitamin D deficiency so the 16% cited isn't an adequate comparator.

We were interested in Vitamin D status closest to diagnosis, which may help guide our clinical practice of screening for Vitamin D in the study population. Therefore, the levels were obtained closest to diagnosis as noted above. We did not collect data on subsequent levels, but this would be an important outcome to clarify in our population for future step.

We agree that reference 32 is not an ideal comparator and updated accordingly in the discussion to note the limitations of the comparison.

6) Much of the dietary data is descriptive in nature without clear statistical analysis but statements are made about certain foods being consumed at rare rates. Is there any strict definition of "rare"?

Since there is no validated dietary tool for pre and post-diagnosis patterns or dietary tool in IBD children, we relied on a practical definition of “rare” when we described intake of “traditionally preserved foods such as pickled mango, kimchi, were rarely consumed”. Most subjects indicated they consume these less than a few times a month.

Minor Concerns:

1) It's not fully clear to me what is meant in the patient population section that the charts of cases were reviewed by "a gastroenterologist and a study team member" (i.e. whether 2 people reviewed it or whether it was a study team member that was a gastroenterologist).

We updated the methods section to say two people reviewed it independently.

2) It's not fully clear to me why international second opinions were excluded but other second opinions remained in (as their phenotype might be less accurate)

We opted to exclude international second opinions but to include second opinions from within the United States because many of the second opinions end up establishing care at our center after their initial visit and would be reflective of the patient population followed at our center. We acknowledge that second opinions may affect our phenotype data as they might be skewed in severity and other clinical characteristics. These limitations are associated with the single-center nature of this study and therefore larger multiple center studies such as that using ICN registry would be needed to fully understand the phenotype of these patients.

3) When the age at diagnosis is discussed for the Asian patients with IC, it is stated that it is lower, although it's not clear what comparative group is being used. I suspect this is "lower than Asian UC cohort and Asian CD cohort" but that should be clarified. Similarly, in that section, there is the statement of a large percentage of patients diagnosed before 6 years of age but not clear if this is the Asian IBD group or Asian IC group (or another group).

We updated the result section to clarify that it is in comparison with the Asian UC and CD subgroups. We also updated the large percentage of patients referred to the Asian IBD cohort.

4) When zip code data was analyzed and 92% of patients were reported to reside "locally", it's not clear what is meant: within state? within several states? by geographic distance?

We updated the manuscript to include PA where our center is located, as well as the nearby states our center reaches (NY, NJ, DE).

5) In the Asian UC cohort, they describe that the "highest proportion of the patients have pancolitis". I suspect this is meant to be the majority of patients have pancolitis rather than this cohort has a higher percentage of pancolitis (since Table 1 suggests that is not the case).

We updated the wording to state “majority of patients have pancolitis”.

6) The phenotype of the South Asian patients seems to be quite similar. It seems that this section could be shortened and that table be moved in as a Supplementary Table. The wording around the perianal Crohn's involvement gives the impression that this is different than the overall Asian cohort but it appears that it is quite similar.

We agree. We moved the table on South Asian patients to Supplemental materials. We updated the wording around perianal Crohn’s involvement to give appropriate impression.

7) It might be helpful to also state what percentage of patients have EITHER a first or second degree relative with IBD. It's not clear whether the 16 out of 79 patients that had a second degree family member with IBD did NOT have a first degree family member with IBD.

We added a sentence to include the percentage of Asian patients with any family history of IBD.

8) There appears to be a word missing in the paragraph in the Discussion section regarding Vitamin D status that starts with "Furthermore"

Thank you, word added to clarify the sentence.

---

## [Decision Letter · Decision Letter 1]

5 Feb 2023

CLINICAL AND LIFESTYLE PATTERNS IN ASIAN CHILDREN WITH INFLAMMATORY BOWEL DISEASE IN THE U.S.

PONE-D-22-24143R1

Dear Dr. Albenberg,

We’re pleased to inform you that your manuscript has been judged scientifically suitable for publication and will be formally accepted for publication once it meets all outstanding technical requirements.

Kind regards,

Valérie Pittet, PhD

Academic Editor

PLOS ONE

Additional Editor Comments (optional):

Reviewers' comments:

Reviewer's Responses to Questions

**Comments to the Author**

1. If the authors have adequately addressed your comments raised in a previous round of review and you feel that this manuscript is now acceptable for publication, you may indicate that here to bypass the “Comments to the Author” section, enter your conflict of interest statement in the “Confidential to Editor” section, and submit your "Accept" recommendation.

Reviewer #1: All comments have been addressed

Reviewer #2: All comments have been addressed

2. Is the manuscript technically sound, and do the data support the conclusions?

Reviewer #1: Yes

Reviewer #2: Yes

3. Has the statistical analysis been performed appropriately and rigorously? 

Reviewer #1: Yes

Reviewer #2: Yes

4. Have the authors made all data underlying the findings in their manuscript fully available?

Reviewer #1: Yes

Reviewer #2: Yes

5. Is the manuscript presented in an intelligible fashion and written in standard English?

Reviewer #1: Yes

Reviewer #2: Yes

6. Review Comments to the Author

Reviewer #1: I have no additional comments and all my concerns were addressed. The manuscript is ready for publication if other reviewers concerns are addressed.

Reviewer #2: The authors have made improvements to the manuscript based on the comments that I had submitted and I have no further concerns.

7. PLOS authors have the option to publish the peer review history of their article (what does this mean?). If published, this will include your full peer review and any attached files.

Reviewer #1: No

Reviewer #2: No

---

## [Editor Report · Acceptance letter]

10 Mar 2023

PONE-D-22-24143R1 

Clinical and lifestyle patterns in Asian children with inflammatory bowel disease in the U.S. 

Dear Dr. Albenberg:

I'm pleased to inform you that your manuscript has been deemed suitable for publication in PLOS ONE. Congratulations! Your manuscript is now with our production department. 

Kind regards, 

on behalf of

PD Dr. Valérie Pittet 

Academic Editor

PLOS ONE